# Conditional Flow Variational Autoencoders for Structured Sequence Prediction

## Abstract

Prediction of future states of the environment and interacting agents is a key competence required for autonomous agents to operate successfully in the real world. Prior work for structured sequence prediction based on latent variable models imposes priors with limited expressiveness or are difficult to optimize e.g. determining the number of Gaussian mixture components which makes it challenging to fully capture the multi-modality of the distribution of the future states. In this work, we introduce *Conditional Flow Variational Autoencoders (CF-VAE)* using our novel conditional normalizing flow based prior to capture complex multi-modal conditional distributions for effective structured sequence prediction. Moreover, we propose two novel regularization schemes which stabilizes training and deals with posterior collapse for stable training and better fit to the target data distribution. Our experiments on three multi-modal structured sequence prediction datasets – MNIST Sequences, Stanford Drone and HighD – show that the proposed method obtains state of art results across different evaluation metrics.

## 1 Introduction

Anticipating future states of the environment is a key competence necessary for the success of autonomous agents. In complex real world environments, the future is highly uncertain. Therefore, structured predictions, one to many mappings of the likely future states of the world, are important. In many scenarios, these tasks can be cast as sequence prediction problems. Particularly, Conditional Variational Autoencoders (CVAE) (Sohn et al., 2015; Bayer & Osendorfer, 2014; Chung et al., 2015) have been very successful – from prediction of pedestrians trajectories (Lee et al., 2017; Bhattacharyya et al., 2018; Pajouheshgar & Lampert, 2018) to outcomes of robotic actions (Babaeizadeh et al., 2018). The distribution of future sequences is diverse and highly multi-modal. CVAEs model diverse futures by factorizing the distribution of future states using a set of latent variables which are mapped to likely future states. However, CVAEs assume a standard Gaussian prior on the latent variables which induces a strong model bias (Hoffman & Johnson, 2016; Tomczak & Welling, 2018) which makes it challenging to capture multi-modal distributions. This also leads to missing modes due to posterior collapse (Bowman et al., 2016; Razavi et al., 2019).

Recent work (Tomczak & Welling, 2018; Wang et al., 2017; Gu et al., 2018) has therefore focused on more complex Gaussian mixture based priors. Gaussian mixtures still have limited expressiveness and optimization suffers from complications e.g. determining the number of mixture components. Normalizing flows are more expressive and enable the modelling of complex multi-modal priors. Recent work on flow based priors (Chen et al., 2017; Ziegler & Rush, 2019), have focused only on the unconditional (plain VAE) case. However, this not sufficient for CVAEs because in the conditional case the complexity of the distributions are highly dependent on the condition.

In this work, 1. We propose *Conditional Flow Variational Autoencoders (CF-VAE)* based on novel conditional normalizing flow based priors In order to model complex multi-modal conditional distributions over sequences. In Figure 1, we show example predictions of MNIST handwriting stroke of our CF-VAE. We observe that, given a starting stroke, our CF-VAE model with data dependent normalizing flow based latent prior captures the two main modes of the conditional distribution – i.e. 1 and 8 – while CVAEs with fixed uni-modal Gaussian prior predictions have limited diversity. 2. We propose a regularization scheme that stabilizes the optimization of the evidence lower bound and leads to better fit to the target data distribution. 3. We leverage our conditional flow prior to deal with posterior collapse which causes standard CVAEs to ignore modes in sequence prediction tasks. 4. Finally, our method outperforms the state of the art on three structured sequence prediction tasks – handwriting stroke prediction on MNIST, trajectory prediction on Stanford Drone and HighD.

| Latent Prior | Clustered Predictions | Latent Prior | Clustered Predictions |
|---|---|---|---|

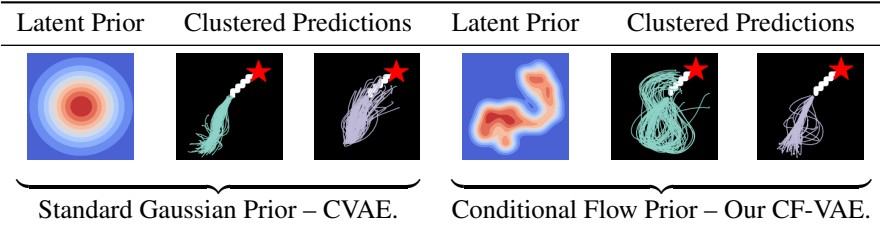

Standard Gaussian Prior – CVAE.          Conditional Flow Prior – Our CF-VAE.

Figure 1: Clustered stroke predictions on MNIST sequences. Our multi-modal Conditional Normalizing Flow based prior (right) enables our regularized CF-VAE to capture the two modes of the conditional distribution, while predictions with uni-modal Gaussian prior (left) have limited diversity. Note, our 64D CF-VAE latent distribution is (approximately) projected to 2D using tSNE and KDE.

## 2 RELATED WORK

**Normalizing Flows.** Normalizing flows are a powerful class of density estimation methods with exact inference. (Dinh et al., 2015) introduced affine normalizing flows with triangular Jacobians. (Dinh et al., 2017) extend flows with masked convolutions which allow for complex (non-autoregessive) dependence between the dimensions. In (Kingma & Dhariwal, 2018), $1 \times 1$ convolutions were proposed for improved image generation compared to (Dinh et al., 2017). In (Huang et al., 2018) normalizing flows are auto-regressive and (Behrmann et al., 2019) extend it to ResNet. (Lu & Huang, 2019) extended normalizing flows to model conditional distributions. Here, we propose conditional normalizing flows to learn conditional priors for variational latent models.

**Variational Autoencoders.** The original variational autoencoder (Kingma & Welling, 2014) used uni-modal Gaussian prior and posterior distributions. Thereafter, two lines of work have focused on developing either more expressive prior or posterior distributions. Rezende & Mohamed (2015) propose normalizing flows to model complex posterior distributions. Kingma et al. (2016); Tomczak & Welling (2016); Berg et al. (2018) present more complex inverse autoregressive flows, householder and Sylvester normalizing flow based posteriors. Here, we focus on the orthogonal direction of more expressive priors and the above approaches are compatible with our approach.

Recent work which focus more expressive priors include (Nalisnick & Smyth, 2017) which proposes a Dirichlet process prior and (Goyal et al., 2017) which proposes a nested Chinese restaurant process prior. However, these methods require sophisticated learning methods. In contrast, (Tomczak & Welling, 2018) proposes a mixture of Gaussians based prior (with fixed number of components) which is easier to train and shows promising results on some image generation tasks. (Chen et al., 2017), proposes a inverse autoregressive flow based prior which leads to improvements in complex image generation tasks like CIFAR-10. (Ziegler & Rush, 2019) proposes a prior for VAE based text generation using complex non-linear flows which allows for complex multi-modal priors. While these works focus on unconditional priors, we aim to develop more expressive conditional priors.

**Posterior Collapse.** Posterior collapse arises when the latent posterior does not encode useful information. Most prior work (Yang et al., 2017; Dieng et al., 2019; Higgins et al., 2017) concentrate on unconditional VAEs and modify the training objective – the KL divergence term is annealed to prevent collapse to the prior. Liu et al. (2019) extends KL annealing to CVAEs. However, KL annealing does not optimize a true lower bound of the ELBO for most of training. Zhao et al. (2017) also modifies the objective to choose the model with the maximal rate. Razavi et al. (2019) propose anti-causal sequential priors for text modelling tasks. Bowman et al. (2016); Gulrajani et al. (2017) proposes to weaken the decoder so that the latent variables cannot be ignored, however only unconditional VAEs are considered. Wang & Wang (2019) shows the advantage of normalizing flow based posteriors for preventing posterior collapse. In contrast, we study for the first time posterior collapse in conditional models on datasets with minor modes.

**Structured Sequence Prediction.** Helbing & Molnar (1995); Robicquet et al. (2016); Alahi et al. (2016); Gupta et al. (2018); Zhao et al. (2019); Sadeghian et al. (2019) consider the problem of traffic participant trajectory prediction in a social context. Notably, (Gupta et al., 2018; Zhao et al., 2019; Sadeghian et al., 2019) use generative adversarial networks to generate socially compliant trajectories. However, the predictions are uni-modal. Starting from Bayer & Osendorfer (2014); Chung et al.

(2015), more recently Lee et al. (2017); Bhattacharyya et al. (2018); Rhinehart et al. (2018); Deo & Trivedi (2019); Pajouheshgar & Lampert (2018) considers structured (one to many) predictions using – a CVAE, improved CVAE training, pushforward policies for vehicle ego-motion prediction, motion planning, spatio-temporal convolutional network respectively. Kumar et al. (2019) proposes a normalizing flow based model for video sequence prediction, however the sequences considered have very limited diversity compared to the trajectory prediction tasks considered here. Here, we focus on improving structured predictions using conditional normalizing flows based priors.

## 3 CONDITIONAL FLOW VARIATIONAL AUTOENCODER (CF-VAE)

Our Conditional Flow Variational Autoencoder is based on the conditional variational autoencoder (Sohn et al., 2015) which is a deep directed graphical model for modeling conditional data distributions $p_\theta(\mathbf{y}|\mathbf{x})$. Here, x is the sequence up to time $t$, $x = [x^1, \cdots, x^t]$ and y is the sequence to be predicted up to time $T$, $y = [y^{t+1}, \cdots, y^T]$. CVAEs factorize the conditional distribution using latent variables z. In detail, $p_\theta(\mathbf{y}|\mathbf{x}) = \int p_\theta(\mathbf{y}|\mathbf{z}, \mathbf{x})p(\mathbf{z}|\mathbf{x})d\mathbf{z}$, where $p(\mathbf{z}|\mathbf{x})$ is the prior on the latent variables. During training, amortized variational inference is used and the posterior distribution $q_\phi(\mathbf{z}|\mathbf{x}, \mathbf{y})$ is learnt using a recognition network. The ELBO is maximized, given by,

$$\log(p_\theta(\mathbf{y}|\mathbf{x})) \geq \mathbb{E}_{q_\phi(\mathbf{z}|\mathbf{x}, \mathbf{y})} \log(p_\theta(\mathbf{y}|\mathbf{z}, \mathbf{x})) - D_{\text{KL}}(q_\phi(\mathbf{z}|\mathbf{x}, \mathbf{y})||p(\mathbf{z}|\mathbf{x})). \tag{1}$$

In practice, to simplify learning, simple unconditional standard Gaussian priors are used (Sohn et al., 2015). However, the complexity e.g. the number of modes of the target distributions $p_\theta(\mathbf{y}|\mathbf{x})$, is highly dependent upon the condition $x$. An unconditional prior demands identical latent distributions irrespective complexity of the target conditional distribution – a very strong constraint on the recognition network. Moreover, the latent variables cannot encode any conditioning information and this leaves the burden of learning the dependence on the condition completely on the decoder.

Furthermore, on complex conditional multi-modal data, Gaussian priors have been shown to induce a strong model bias (Tomczak & Welling, 2016; Ziegler & Rush, 2019). It becomes increasingly difficult to map complex multi-modal distributions to uni-modal Gaussian distributions, further complicated by the sensitivity of the RNNs encoder/decoders to subtle variations in the hidden states (Bowman et al., 2016). Moreover, the standard closed form estimate of the KL-divergence pushes the encoded latent distributions to the mean of the Gaussian leading to latent variable collapse (Wang et al., 2017; Gu et al., 2018) while discriminator based approaches (Tolstikhin et al., 2017) lead to underestimates of the KL-divergence (Rosca et al., 2017).

Therefore, we propose conditional priors based on conditional normalizing flows to enable the latent variables to encode conditional information and allow for complex multi-modal latent representations. Next, we introduce our new conditional non-linear normalizing flows followed by our regularized Conditional Flow Variational Autoencoder (CF-VAE) formulation.

### 3.1 CONDITIONAL NORMALIZING FLOWS

Recently, normalizing flow (Tabak et al., 2010; Dinh et al., 2015) based priors for VAEs have been proposed (Chen et al., 2017; Ziegler & Rush, 2019). Normalizing flows allows for complex priors by transforming a simple base density e.g. standard Gaussian to a complex multi-modal density through a series of $n$ layers of invertible transformations $f_i$,

$$\epsilon \xleftrightarrow{f_1} \mathbf{h}_1 \xleftrightarrow{f_2} \mathbf{h}_2 \cdots \xleftrightarrow{f_n} \mathbf{z}. \tag{2}$$

However, such flows cannot model conditional priors. In contrast to prior work, we utilize conditional normalizing flows to model complex conditional priors. Conditional normalizing flows also consists of a series of $n$ layers of invertible transformations $f_i$ (with parameters $\psi$), however we modify the transformations $f_i$ such that they are dependent on the condition x,

$$\epsilon|\mathbf{x} \xleftrightarrow{f_1|\mathbf{x}} \mathbf{h}_1|\mathbf{x} \xleftrightarrow{f_2|\mathbf{x}} \mathbf{h}_2|\mathbf{x} \cdots \xleftrightarrow{f_n|\mathbf{x}} \mathbf{z}|\mathbf{x}. \tag{3}$$

Further, in contrast to prior work (Lu & Huang, 2019; Atanov et al., 2019; Ardizzone et al., 2019) which use affine flows ($f_i$), we build upon (Ziegler & Rush, 2019) and introduce conditional non-linear normalizing flows with split coupling. Split couplings ensure invertibility by applying a flow

layer $f_i$ on only half of the dimensions at a time. To compute (5), we split the dimensions $z^D$ of the latent variable into halves, $z^L = \{1, \cdots, D/2\}$ and $z^R = \{D/2, \cdots, d\}$ at each invertible layer $f_i$. Our transformation takes the following form for each dimension $z^j$ alternatively from $z^L$ or $z^R$,

$$f_i^{-1}(z^j|z^R, x) = \epsilon^j = a(z^R, x) + b(z^R, x) \times z^j + \frac{c(z^R, x)}{1 + \left(d(z^R, x) \times z^j + g\left(z^R, x\right)\right)^2}. \quad (4)$$

where, $z^j \in z^L$. Details of the forward (generating) operation $f_i$ are in Appendix A. To ensure that the generated prior distribution is conditioned on x, in (4) and in the corresponding forward operation $f_i$, the coefficients $\{a, b, c, d, g\} \in \mathbb{R}$ are functions of both the other half of the dimensions of z *and* the condition x (unlike Ziegler & Rush (2019)). Finally, due to the expressive power of our conditional non-linear normalizing flows, simple spherical Gaussians base distributions were sufficient.

## 3.2 VARIATIONAL INFERENCE USING CONDITIONAL NORMALIZING FLOWS BASED PRIORS

Here, we derive the ELBO (1) for our regularized CF-VAE with our conditional flow based prior. In case of the standard CVAE with the Gaussian prior, the KL divergence term in the ELBO has a simple closed form expression. In case of our conditional flow based prior, we can use the change of variables formula to compute the KL divergence. In detail, given the base density $p(\epsilon|x)$ and the Jacobian $J_i$ of each layer $i$ of the transformation, the log-likelihood of the latent variable z under the prior can be expressed using the change of variables formula,

$$\log(p_\psi(z|x)) = \log(p(\epsilon|x)) + \sum_{i=1}^{n} \log(|\det J_i|). \quad (5)$$

This change of variables allows us to evaluate the likelihood of latent variable z over the base distribution instead of the complex conditional prior and to express the KL divergence as,

$$-D_{\text{KL}}(q_\phi(z|x, y)||p_\psi(z|x)) = -\mathbb{E}_{q_\phi(z|x,y)} \log(q_\phi(z|x, y)) + \mathbb{E}_{q_\phi(z|x,y)} \log(p_\psi(z|x))$$

$$= \mathcal{H}(q_\phi) + \mathbb{E}_{q_\phi(z|x,y)} \log(p(\epsilon|x)) + \sum_{i=1}^{n} \log(|\det J_i|). \quad (6)$$

where, $\mathcal{H}(q_\phi)$ is the entropy of the variational distribution. Therefore, the ELBO can be expressed as,

$$\log(p_\theta(y|x)) \geq \mathbb{E}_{q_\phi(z|x,y)} \log(p_\theta(y|z, x)) + \mathcal{H}(q_\phi) + \mathbb{E}_{q_\phi(z|x,y)} \log(p(\epsilon|x)) + \sum_{i=1}^{n} \log(|\det J_i|) \quad (7)$$

To learn complex conditional priors, we alternately optimize both the variational posterior distribution $q_\phi(z|x, y)$ and the conditional prior $p_\psi(z|x)$ in (7). This would allow the variational posterior $q_\theta$ to match the conditional prior and vice-versa so that the ELBO (7) is maximized. However, in practice we observe instabilities during training and posterior collapse. Next, we introduce our novel regularization schemes to deal with both these problems.

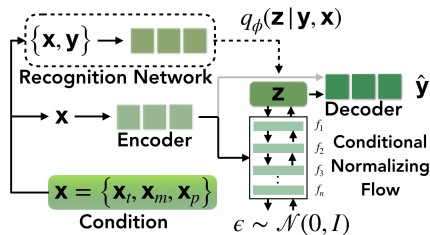

Figure 2: CF-VAE. The decoder is regularized by removing conditioning (grey arrow) to prevent posterior collapse.

**Posterior Regularization for Stability (pR).** The entropy and the $\log$-Jacobian of the CF-VAE objective (7) are at odds with each other. The $\log$-Jacobian favours the contraction of the base density. Therefore, $\log$-Jacobian at the right of (7) is maximized when the conditional flow maps the base distribution ($\epsilon \leftrightarrow z$ in Figure 2) to a low entropy conditional prior and thus a low entropy variational distribution $q_\phi(z|x, y)$. Therefore, in practice we observe instabilities during training. We observe that either the entropy or the $\log$-Jacobian term dominates and the data log-likelihood is fully or partially ignored. Therefore, we regularize the posterior $q_\phi(z|x, y)$ by fixing the variance to C. This leads to a constant entropy term which in turn bounds the maximum possible amount of contraction, thus upper bounding the $\log$-Jacobian. This encourages our model to concentrate on explaining the data and leads better fit to the target data distribution. Note that, although $q_\phi(z|x, y)$

has fixed variance, this does not significantly effect sample quality as the marginal $q_\phi(z|x)$ can be arbitrarily complex due to our conditional flow prior. Moreover, we observe that the LSTM based decoders employed demonstrate robust performance across a wide range of values $C = [0.05, 0.25]$.

**Condition Regularization for Posterior Collapse (cR).** We observe missing modes when the target conditional data distribution has a major mode(s) and one or more minor modes (corresponding to rare events). This is because the condition x on the decoder is already enough to model the main mode(s). If the cost of ignoring the minor modes is out-weighed by the cost of encoding a more complex latent distribution reflecting all modes, the minor modes and the latent variables are ignored. We propose a regularization scheme by removing the additional conditioning x on the decoder, when the dataset in question has a dominating mode(s). This enabled by our conditional flow prior, which ensures that conditioning information is encoded in the latent space and $p_\theta(y|z)$ can match $p_\theta(y|x, z)$. Leading to a simpler factorization, $p_\theta(y|x) = \int p_\theta(y|z)p_\psi(z|x)dz$. Equivalently, this ensures that the latent variable z cannot be ignored by the CF-VAE and thus must encode useful information. Note that this regularization scheme is only possible due to our conditional prior, the unconditional Gaussian prior of CVAE would always need to condition the decoder.

The parallel work of Klushyn et al. (2019) also proposes a similar regularization scheme. However, we employ this regularization to deal with posterior collapse only in case of distributions with dominant modes. We also provide a more detailed analysis of their proposed prior in Appendix E.

Finally, we discuss the integration of diverse sources of contextual information into the conditional prior $p_\psi(z|x)$ for even richer conditional latent distributions of our regularized CF-VAE.

## 3.3 Conditioning Priors on Contextual Information

For prediction tasks, it is often crucial to integrate sources of contextual information e.g. past trajectories or environmental information for accurate predictions. As these sources are heterogeneous, we employ source specific networks to extract fixed length vectors from each source.

**Past Trajectory.** We encode the past trajectories using a LSTM to an fixed length vector $x_t$. For efficiency we share the condition encoder between the conditional flow and the CF-VAE decoder.

**Environmental Map.** We use a CNN to encode environmental information to a set of region specific feature vectors. We apply attention conditioned on the past trajectory to extract a fixed length conditioning vector $x_m$, such that $x_m$ contains information relevant to the future trajectory.

**Interacting Agents.** To encode information of interacting traffic participants/agents, we build on Deo & Trivedi (2018) and propose a fully convolutional social pooling layer. We aggregate information of interacting agents using a grid overlayed on the environment. This grid is represented using a tensor, where the past trajectory information of traffic participants are aggregated into the tensor indexed corresponding to the grid in the environment. In Deo & Trivedi (2018) past trajectory information is aggregated using a LSTM. We aggregate the past trajectory information into the tensor using $1 \times 1$ convolutions as it allows for stable learning and is computationally efficient. Finally, we apply several layers of $k \times k$ convolutions to capture interaction aware contextual features $x_p$ of traffic participants in the scene.

Due to the expressive power of our conditional non-linear normalizing flows, simple concatenation into a single vector $x = \{x_t, x_m, x_t\}$ was sufficient to learn powerful conditional priors.

## 4 Experiments

We evaluate our CF-VAE on three popular and highly multi-modal sequence prediction datasets. We begin with a description of our evaluation metrics and model architecture.

**Evaluation Metrics.** In line with prior work (Lee et al., 2017; Bhattacharyya et al., 2018; Pajouheshgar & Lampert, 2018; Deo & Trivedi, 2019; Bhattacharyya et al., 2019), we use the negative conditional $\log$-likelihood (-CLL) and mean Euclidean distances of the oracle Top $n\%$ of $N$ predictions. The oracle Top $n\%$ metric measures not only the coverage of all modes but also discourages random guessing for a reasonably large value of $n$ (e.g. $n = 10\%$). This is because, a model can only improve this metric by moving randomly guessed samples from an overestimated mode to the correct modes (detailed analysis in Appendix F).

| Condition | BMS-CVAE Modes (Bhattacharyya et al., 2018) | | | Our CF-VAE Modes | | | Our CF-VAE Prior |

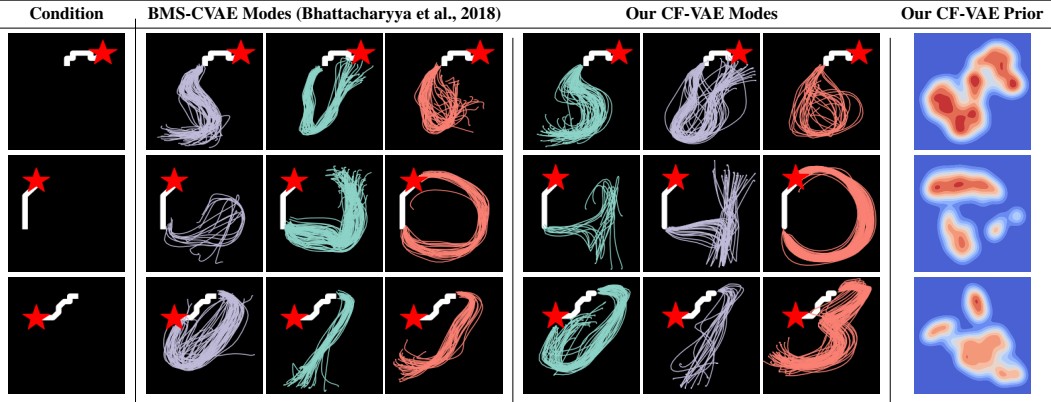

Figure 3: Random samples clustered using k-means. The number of clusters is set manually to the number of expected digits. The corresponding priors of our CF-VAE + pR on the right. Note, our 64D CF-VAE latent distribution is (approximately) projected to 2D using tSNE and KDE.

**Conditional Flow Model Architecture.** Our conditional flow prior consists of 16 layers of conditional non-linear flows with split coupling. Increasing the number of conditional non-linear flows generally led to "over-fitting" on the training latent distribution.

### 4.1 MNIST SEQUENCES

The MNIST Sequence dataset (D. De Jong, 2016) consists of sequences of handwriting strokes of the MNIST digits. The state-of-the-art approach is the "Best-of-Many"-CVAE (Bhattacharyya et al., 2018) with a Gaussian prior. We follow the evaluation protocol of Bhattacharyya et al. (2018) and predict the complete stroke given the first ten steps. We also compare with, 1. A standard CVAE with uni-modal Gaussian prior; 2. A CVAE with a data dependent conditional mixture of Gaussians (MoG) prior; 3. A CF-VAE without any regularization ; 4. A CF-VAE without the conditional non-linear flow layers (CF-VAE-*Affine*, replaced with affine flows (Lu & Huang, 2019; Atanov et al., 2019)). We also experiment with a conditional MoG prior (see Appendix D and E). We use the same model architecture (Bhattacharyya et al., 2018) across all baselines.

We report the results in Table 1. We see that our CF-VAE with posterior regularization (pR) performs best. It has a performance advantage of over 20% against the state of the art BMS-CVAE. We see that without regularization (pR) (C = 0.2) there is a 40% drop in performance, highlighting the effectiveness of our proposed regularization scheme. We further illustrate the modes captured and the learnt multi-modal conditional flow priors in Figure 3. We do not use condition regularization here (cR) as we do not observe posterior collapse. In contrast, the BMS-CVAE is unable to fully capture all modes – its predictions are pushed to the

| Method | -CLL ↓ |
|---|---|
| CVAE (Sohn et al., 2015) | 96.4 |
| BMS-CVAE (Bhattacharyya et al., 2018) | 95.6 |
| CVAE + *increased capacity* (Ours) | 94.5 |
| CVAE + *conditional prior* (Ours) | 88.9 |
| MoG-CVAE, $M = 3$ | 84.6 |
| CF-VAE - *no regularization* (Ours) | 104.3 |
| CF-VAE - *Affine* + pR, C = 0.2 (Ours) | 77.2 |
| CF-VAE + pR, C = 0.2 (Ours) | **74.9** |

Table 1: Evaluation on MNIST Sequences.

mean due to the strong model bias induced by the Gaussian prior. The results improve considerably with the multi-modal MoG prior ($M = 3$ components work best). We also experiment with optimizing the standard CVAE architecture. This improves performance only slightly (after increasing LSTM encoder/decoder units to 256 from 48, increasing the number of layers did not help). Moreover, our experiments with a conditional (MoG) AAE/WAE (Gu et al., 2018) based baseline did not improve performance beyond the standard (MoG) CVAE, because the discriminator based KL estimate tends to be an underestimate (Rosca et al., 2017). This illustrates that in practice it is difficult to map highly multi-modal sequences to a Gaussian prior and highlights the need of a data-dependent multi-modal priors. Our CF-VAE still significantly outperforms the MoG-CVAE as normalizing flows are better at learning complex multi-modal distributions (Kingma & Dhariwal, 2018). We also see that affine conditional flow based priors leads to a drop in performance (77.2 vs 74.9 CLL) illustrating the advantage of our non-linear conditional flows.

| Method | Visual | Error @ 1sec | Error @ 2sec | Error @ 3sec | Error @ 4sec | -CLL ↓ |
|---|---|---|---|---|---|---|
| "Shotgun" (Top 10%) (Pajouheshgar & Lampert, 2018) | None | 0.7 | 1.7 | 3.0 | 4.5 | 91.6 |
| DESIRE-SI-IT4 (Top 10%) (Lee et al., 2017) | RGB | 1.2 | 2.3 | 3.4 | 5.3 | x |
| STCNN (Top 10%) (Pajouheshgar & Lampert, 2018) | RGB | 1.2 | 2.1 | 3.3 | 4.6 | x |
| BMS-CVAE (Top 10%) (Bhattacharyya et al., 2018) | RGB | 0.8 | 1.7 | 3.1 | 4.6 | 126.6 |
| MoG-CVAE, $M = 3$ (Top 10%) | None | 0.8 | 1.7 | 2.7 | 3.9 | 86.1 |
| CF-VAE - *no regularization* (Ours, Top 10%) | None | 0.9 | 1.9 | 3.3 | 4.7 | 96.2 |
| CF-VAE + pR, $C = 0.2$ (Ours, Top 10%) | None | **0.7** | **1.5** | 2.5 | 3.6 | 84.6 |
| CF-VAE + pR, $C = 0.2$ (Ours, Top 10%) | RGB | **0.7** | **1.5** | **2.4** | **3.5** | **84.1** |

Table 2: Five fold cross validation on the Stanford Drone dataset. Euclidean error at ($1/5$) resolution.

## 4.2 STANFORD DRONE

| Sampled Predictions | Latent Prior | Sampled Predictions | Latent Prior | Sampled Predictions | Latent Prior |
|---|---|---|---|---|---|

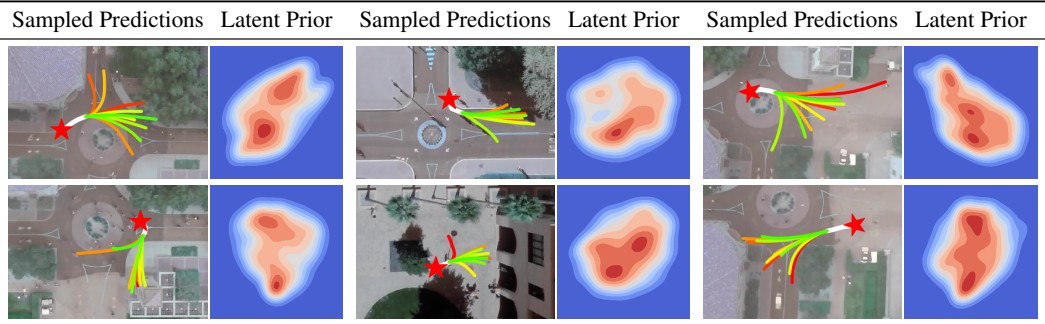

Figure 4: Randomly sampled predictions of our CF-VAE + pR model on the Stanford Drone. We observe that our prediction are highly multi-modal and is reflected by the Conditional Flow Priors. Note, our 64D CF-VAE latent distribution is (approximatly) projected to 2D using tSNE and KDE.

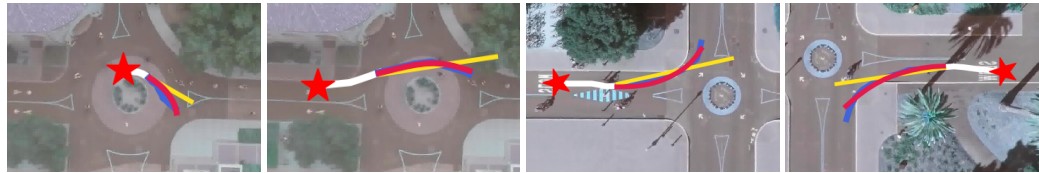

Figure 5: Comparison of our CF-VAE + pR (Red) and the "Shoutgun" baseline (Yellow) of (Pajouheshgar & Lampert, 2018), Groundtruth (Blue). Initial conditioning trajectory in white. Our CF-VAE not only learns to capture the correct modes but also generates more fine-grained predictions.

The Stanford Drone dataset (Robicquet et al., 2016) consists of multi-model trajectories of traffic participant e.g. pedestrians, bicyclists, cars captured from a drone. Prior works follow two different evaluation protocols, 1. (Lee et al., 2017; Bhattacharyya et al., 2018; Pajouheshgar & Lampert, 2018) use 5 fold cross validation, 2. (Robicquet et al., 2016; Sadeghian et al., 2018; 2019; Deo & Trivedi, 2019) use a single split. We evaluate using the first protocol in Table 2 and the second in Table 3.

Additionally, Pajouheshgar & Lampert (2018) suggest a "Shotgun" baseline. This baseline extrapolates the trajectory from the last known position and orientation in 10 different ways – 5 orientations: ($0°$, $\pm 8°$, $\pm 15°$) and 5 velocities: None or exponentially weighted over the past with coefficients ($0$, $0.3$, $0.7$, $1.0$). This baseline obtains results at par with the state-of-the-art because it a good template which covers the most likely possible futures (modes) for traffic participant motion

| Method | mADE ↓ | mFDE ↓ |
|---|---|---|
| SocialGAN (Gupta et al., 2018) | 27.2 | 41.4 |
| MATF GAN (Zhao et al., 2019) | 22.5 | 33.5 |
| SoPhie (Sadeghian et al., 2019) | 16.2 | 29.3 |
| Goal Prediction (Deo & Trivedi, 2019) | 15.7 | 28.1 |
| CF-VAE + pR, C = 0.2 (Ours) | **12.6** | **22.3** |

Table 3: Evaluation on the Stanford Drone dataset on a single split (see also Table 2).

in this dataset. We report the results using 5 fold cross validation in Table 2. We additionally compare to a mixture of Gaussians prior (Appendix D). We use the same model architecture as in Bhattacharyya et al. (2018) and a CNN encoder with attention to extract features from the last observed RGB image (Appendix C). These visual features serve as additional conditioning ($x_m$) to our Conditional Flow model. We see that our CF-VAE model with RGB input and posterior

regularization (pR) performs best – outperforming the state-of-art "Shotgun" and BMS-CVAE by over 20% (Error @ 4sec). We see that our conditional flows are able to utilize visual scene (RGB) information to improve performance (3.5 vs 3.6 Error @ 4sec). We also see that the MoG-CVAE and our CF-VAE + pR outperforms the BMS-CVAE, even without visual scene information. This again reinforces our claim that the standard Gaussian prior induces a strong model bias and data dependent multi-modal priors are needed for best performance. The performance advantage of CF-VAE over the MoG-CVAE again illustrates the advantage of normalizing flows at learning complex conditional multi-modal distributions. The performance advantage over the "Shotgun" baseline shows that our CF-VAE + pR not only learns to capture the correct modes but also generates more fine-grained predictions. The qualitative examples in Figure 5 shows that our CF-VAE is better able to capture complex trajectories with sharp turns.

We report results using the single train/test split of (Robicquet et al., 2016; Sadeghian et al., 2018; 2019; Deo & Trivedi, 2019) in Table 3. We use the minimum Average Displacement Error (mADE) and minimum Final Displacement Error (mFDE) metrics as in (Deo & Trivedi, 2019). The minimum is over as set of predictions of size $N$. Although this metric is less robust to random guessing compared to the Top $n\%$ metric, it avoids rewarding random guessing for a small enough value of $N$. We choose $N = 20$ as in (Deo & Trivedi, 2019). Similar to the results with 5 fold cross validation, we observe 20% improvement over the state-of-the-art.

## 4.3 HIGHD

The HighD dataset (Krajewski et al., 2018) consists of vehicle trajectories recorded using a drone over highways. In contrast to other vehicle trajectory datasets e.g. NGSIM it contains minimal false positive trajectory collisions or physically improvable velocities.

The HighD dataset is challenging because lane changes or interactions are rare $\sim 10\%$ of all trajectories. The distribution of future trajectories contain a single main mode (linear continuations) along with several minor modes. Thus, approaches which predict a single mean trajectory (targeting the main mode) are challenging to outperform. In Table 4, we see that the simple Feed Forward (FF) model performs well and the Graph

| Method | Context | ADE ↓ | FDE ↓ | -CLL ↓ |
|---|---|---|---|---|
| Constant Velocity | None | 1.09 | 2.66 | x |
| FF (Diehl et al., 2019) | None | 0.45 | 1.09 | x |
| GAT (Diehl et al., 2019) | Yes | 0.47 | 1.04 | x |
| CVAE (Top 10%) | None | 0.45 | 0.96 | 5.32 |
| CVAE + *Cyclic KL* (Top 10%) | None | 0.38 | 0.80 | 4.80 |
| CF-VAE + pR, (Ours, Top 10%) | None | 0.44 | 0.94 | 4.71 |
| CF-VAE + {pR,cR}, (Ours, Top 10%) | None | 0.30 | 0.57 | 3.64 |
| CF-VAE + {pR,cR}, (Ours, Top 10%) | Yes | **0.29** | **0.55** | **3.42** |

Table 4: Evaluation on the HighD dataset.

Convolutional GAT model of Diehl et al. (2019), which captures interactions, only narrowly outperforms the FF model. This dataset is challenging for CVAE based models as they frequently suffer from posterior collapse when a single mode dominates. This is clearly observed with our CVAE baseline in Table 4. To prevent posterior collapse, we use the cyclic KL annealing scheme proposed in Liu et al. (2019) (using a MoG prior did not help). This already leads to significant improvement over the deterministic FF and GAT baselines. We also observe posterior collapse with our CF-VAE model. Therefore, we regularize by removing additional conditioning (cR). Our CF-VAE + {pR,cR} with condition regularization significantly outperforms the CF-VAE + pR and CVAE baselines (with cyclic KL annealing), demonstrating the effectiveness of our condition regularization scheme (cR) in preventing posterior collapse. The addition of contextual information of interacting traffic participants using our convolutional social pooling network with $1 \times 1$ convolutions significantly improves performance (also see Appendix G), demonstrating the effectiveness of our conditional normalizing flow based priors.

## 5 CONCLUSION

In this work, we presented the first variational model for learning multi-modal conditional data distributions with Conditional Flow based priors – the Conditional Flow Variational Autoencoder (CF-VAE). Furthermore, we propose two novel regularization techniques – posterior regularization (pR) and condition regularization (cR) – which stabilizes training solutions and prevents posterior collapse leading to better fit to the target distribution. This techniques lead to better match to the target distribution. Our experiments on diverse sequence prediction datasets show that our CF-VAE achieves state-of-the-art results across different performance metrics.

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

## APPENDIX A. CONDITIONAL NON-LINEAR NORMALIZING FLOWS

In Subsection 3.1 of the main paper, we describe the inverse operation $f_i^{-1}$ of our non-linear conditional normalizing flows. Here, we describe the forward operation. Note that while the forward operation is necessary to compute the likelihood (3) (in the main paper) during training, the forward operation is necessary to sample from the latent prior distribution of our CF-VAE. The forward operation consists of solving for the roots of the following equation (more details in (Ziegler & Rush, 2019)),

$$
\begin{aligned}
& -bd^2(\epsilon^j)^3 + ((z^j - a)d^2 - 2dgb)(\epsilon^j)^2 \\
& + (2dg(z^j - a) - b(g^2 + 1))\epsilon^j + ((z^j - a)(g^2 + 1) - c) = 0
\end{aligned}
\tag{8}
$$

This equation has one real root which can be found analytically (Holmes). As mentioned in the main paper, note that the coefficients $\{a, b, c, d, g\}$ are also functions of the condition x (unlike (Ziegler & Rush, 2019)).

## APPENDIX B. ADDITIONAL EVALUATION OF CONDITIONAL NON-LINEAR FLOWS

| Given x in, | $p(y|x)$ | Cond Affine Flow | **Our Cond NL Flow** |
|---|---|---|---|

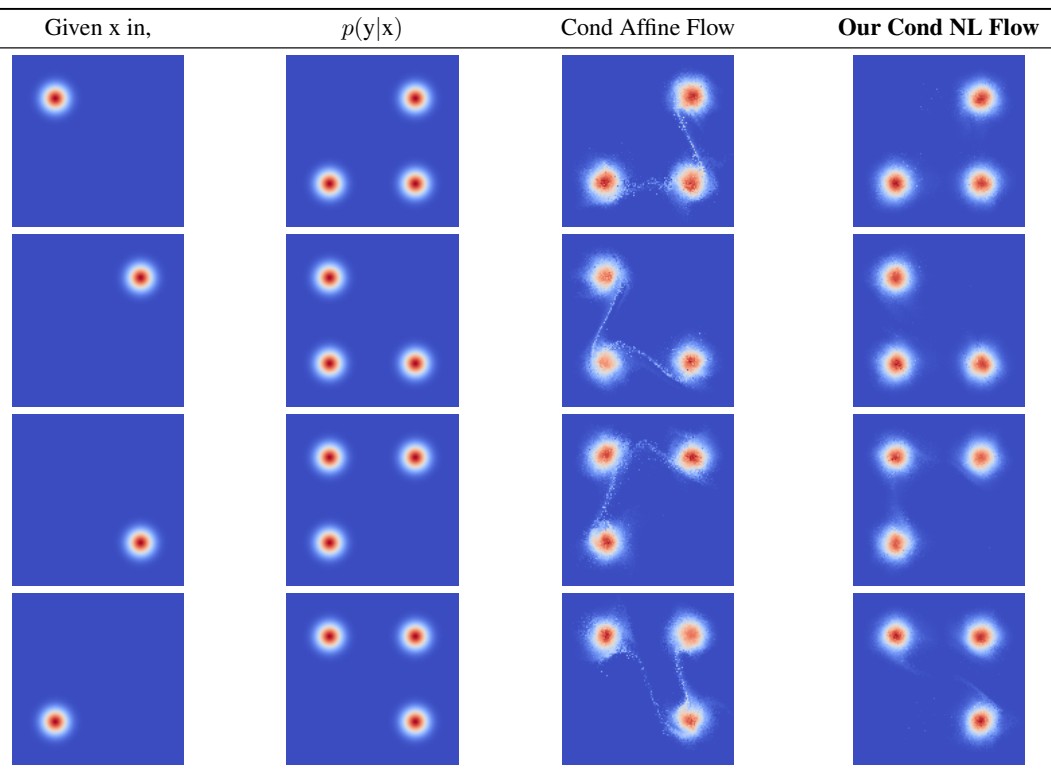

Figure 6: Comparison between conditional affine flows of (Atanov et al., 2019; Lu & Huang, 2019) and our conditional non-linear (Cond NL) flows. We see that the conditional affine flows cannot fully capture multi-modal distributions ("tails" between modes), while our conditional non-linear flows does not have distinctive "tails".

We compare conditional affine flows of (Atanov et al., 2019; Lu & Huang, 2019) and our conditional non-linear (Cond NL) flows in Figure 6 and Figure 7. We plot the conditional distribution $p(y|x)$ and the corresponding condition x in the second and first columns. We use 8 and 16 layers of flow in case of the densities in Figure 6 and Figure 7 respectively. We see that the estimated density by the conditional affine flows of (Atanov et al., 2019; Lu & Huang, 2019) contains distinctive "tails" in case of Figure 6 and discontinuities in case of Figure 7. In comparison our conditional

non-linear flows does not have distinctive "tails" or discontinuities and is able to complex capture the multi-modal distributions better. Note, the "ring"-like distributions in Figure 7 cannot be well captured by more traditional methods like Mixture of Gaussians. We see in Figure 8 that even with 64 mixture components, the learnt density is not smooth in comparison to our conditional non-linear flows. This again demonstrates the advantage of our conditional non-linear flows.

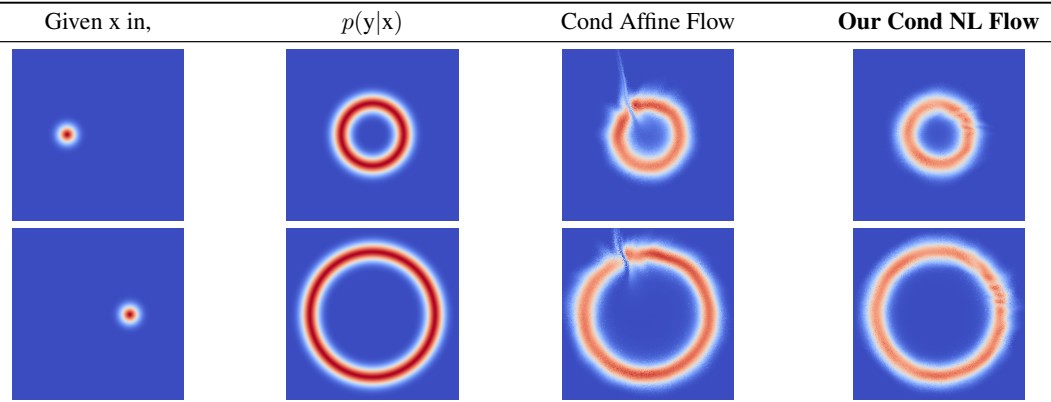

Figure 7: Comparison between conditional affine flows of (Atanov et al., 2019; Lu & Huang, 2019) and our conditional non-linear (Cond NL) flows. We see that the conditional affine flows cannot fully capture "ring"-like conditional distributions (note the discontinuity at the top), while our conditional non-linear flows does not have such discontinuities.

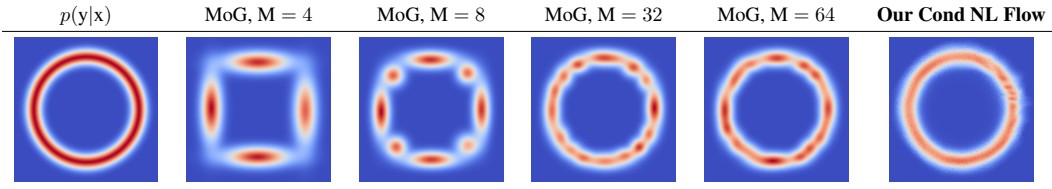

Figure 8: Comparison between our conditional non-linear (Cond NL) flows and a Mixture of Gaussians (MoG) model. We see that even with 64 mixture components, the learnt density is not smooth in comparison to our conditional non-linear flows.

## APPENDIX C. ADDITIONAL DETAILS OF OUR MODEL ARCHITECTURES

Here, we provide details of the model architectures used across the three datasets used in the main paper.

**MNIST Sequences.** We use the same model architecture as in Bhattacharyya et al. (2018). The LSTM condition encoder on the input sequence x, the LSTM recognition network $q_\theta$ and the decoder LSTM network has 48 hidden neurons each. Also as in Bhattacharyya et al. (2018), we use a 64 dimensional latent space.

**Stanford Drone.** Again, we use the same model architecture as in Bhattacharyya et al. (2018) except for the CNN encoder. The LSTM condition encoder on the input sequence x and the decoder LSTM network has 64 hidden neurons each. The LSTM recognition network $q_\theta$ has 128 hidden neurons. Also as in Bhattacharyya et al. (2018), we use a 64 dimensional latent space. Our CNN encoder has 6 convolutional layers of size 32, 64, 128, 256, 512 and 512. We predict the attention weights on the final feature vectors using the encoding of the LSTM condition encoder. The attention weighted feature vectors are passed through a final fully connected layer to obtain the final CNN encoding. Furthermore, we found it helpful to additionally encode the past trajectory as an image (as in (Pajouheshgar & Lampert, 2018)) as provide this as an additional channel to the CNN encoder.

**HighD.** We use the same model architecture with both the CVAE and CF-VAE models. As in the Stanford drone dataset, we use LSTM condition encoder on the input sequence x and the decoder LSTM network with 64 hidden neurons each and the LSTM recognition network $q_\theta$ with 128 hidden neurons. The contextual information of interacting traffic participants are encoded into a spatial grid tensor of size 13×3 (see Section 3.2 of the main paper). We use a CNN with 5 layers of sizes 64, 128, 256, 256 and 256 to extract contextual features.

## APPENDIX D. DETAILS OF THE MIXTURE OF GAUSSIANS (MOG) BASELINE

In the main paper, we include results on the MNIST Sequence and Stanford Drone dataset with a Mixture of Gaussians (MoG) prior. In detail, instead of a normalizing flow, we set the prior to a MoG form,

$$p_\xi(\mathbf{z}|\mathbf{x}) = \sum_{i=1}^{M} p(\mathbf{c}_i|\mathbf{x})\mathcal{N}(\mathbf{z}; \mu_i, \sigma_i|\mathbf{x}). \tag{9}$$

We use a simple feed forward neural network that takes in the condition x (see Section 3.4 of the main paper) and predicts the parameters of the MoG, $\xi = \{\mathbf{c}_1, \mu_1, \sigma_1, \cdots, \mathbf{c}_M, \mu_M, \sigma_M\}$. Note, to ensure a reasonable number of parameters, we consider spherical Gaussians. Similar to (5) in the main paper, the ELBO can be expressed as,

$$\log(p_\theta(\mathbf{y}|\mathbf{x})) \geq \mathbb{E}_{q_\phi(\mathbf{z}|\mathbf{x},\mathbf{y})} \log(p_\theta(\mathbf{y}|\mathbf{z},\mathbf{x})) + \mathcal{H}(q_\phi) + \mathbb{E}_{q_\phi(\mathbf{z}|\mathbf{x},\mathbf{y})} \log(p_\xi(\mathbf{z}|\mathbf{x})). \tag{10}$$

Note that we fix the entropy of the posterior distribution $q_\phi$ for stability

## APPENDIX E. ADDITIONAL EVALUATION ON THE MNIST SEQUENCE DATASET

Here, we perform a comprehensive evaluation using the MoG prior with varying mixture components, a CVAE with unconditional non-linear flow based prior (NL-CVAE), our CF-VAE with Volume-preserving constant Jacobian conditional NICE flows based on Dinh et al. (2015), a CVAE with the conditional VampPrior (CDV) of (Klushyn et al., 2019), our CF-VAE with varying hyper-parameters $C = [0.05, 0.25]$ of our posterior regularization (pR) scheme and finally analyze the effect of our posterior regularization (pR) scheme in detail. We report the results in Table 5.

| Method | -CLL ↓ |
|---|---|
| NL-CVAE | 107.6±1.2 |
| CVAE ($M = 1$) (Sohn et al., 2015) | 96.4±0.2 |
| MoG-CVAE, $M = 2$ | 85.3±0.4 |
| MoG-CVAE, $M = 3$ | 84.6±0.5 |
| MoG-CVAE, $M = 4$ | 85.7±0.4 |
| MoG-CVAE, $M = 5$ | 86.3±0.6 |
| CDV (Klushyn et al., 2019), $M = 12$ | 99.4±0.7 |
| CF-VAE - *NICE* (Ours) | 78.9±0.2 |
| CF-VAE + pR, $C = 0.05$, (Ours) | 75.9±0.5 |
| CF-VAE + pR, $C = 0.10$, (Ours) | 75.4±0.3 |
| CF-VAE + pR, $C = 0.15$, (Ours) | 75.1±0.3 |
| CF-VAE + pR, $C = 0.20$, (Ours) | **74.9**±0.2 |
| CF-VAE + pR, $C = 0.25$, (Ours) | 75.8±0.4 |

Table 5: Evaluation on MNIST Sequences (CLL: lower is better).

**MoG Prior.** As mentioned in the main paper, we see that the MoG-CVAE outperforms the plain CVAE. This again reinforces our claim that the standard Gaussian prior induces a strong model bias. We see that using $M = 3$ components with the variance of the posterior distribution fixed

| Condition | CDV Modes | | | CDV Prior |
|---|---|---|---|---|

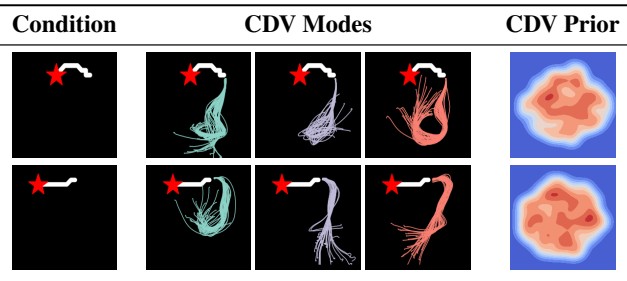

Figure 9: Random samples using the CDV Prior of (Klushyn et al., 2019) clustered using k-means. The number of clusters is set manually to the number of expected digits. The CDV Prior latent distribution on the right. Note, the 64D latent distribution is (approximately) projected to 2D using tSNE and KDE. In comparison to the samples and latent spaces of our CF-VAE (Figure 3) we see that the latent spaces are more simplistic and samples are of poorer quality.

to C = 0.2 leads to the best performance. This is expected as 3 is the most frequent number of possible strokes in the MNIST Sequence dataset. Also note that the results with the MoG prior are also relatively robust across C = [0.05, 0.25] as we learn the variance of the prior (see the section above). Finally, our CF-VAE + pR still significantly outperforms the MoG-CVAE (74.9 vs 84.6). This is expected as normalizing flows are more powerful compared to MoG at learning complex multi-modal distributions (Kingma & Dhariwal, 2018) (also see Figure 8).

**NL-CVAE.** We also see that using an unconditional non-linear flow based prior actually harms performance (107.6 vs 96.4). This is because the latent distribution is highly dependent upon the condition. Therefore, without conditioning information the non-linear conditional flow learns a global representation of the latent space which leads to out-of-distribution samples at prediction time.

**CF-VAE with conditional NICE flows (Dinh et al., 2015).** We have added results with the volume preserving NICE flows in Table 5. We observe that even without our posterior regularization scheme (pR) volume preserving NICE flows (Dinh et al., 2015) performs well – because of the constant Jacobian term. However, our conditional non-linear flows with posterior regularization still perform significantly better (78.9 vs 74.9 -CLL). This is because of the additional expressive power of our conditional non-linear flows combined with the stability offered by our posterior regularization scheme.

**Comparison to Klushyn et al. (2019).** We also perform additional experiments with the conditional VampPrior (CDV) of Klushyn et al. (2019) using $M = 12$ components. Using more components makes training/inference significantly slower in comparison to plain CVAEs, Mog-CVAE ($M = 3$) or our CF-VAE. Furthermore, with $M = 12$ components we observe that it is outperformed by the simpler MoG-CVAE. This is because the mean and variance parameters of the ($M = 12$) components are obtained using the recognition network $q_\phi$. The recognition network $q_\phi$ has to learn to both reconstruct the data and maintain a latent space representative of full conditional data distribution $p(\mathbf{y}|\mathbf{x})$. These objectives are at odds with each other. In practice, we find that this leads to simplistic latent spaces along with lower overall data log-likelihood in comparison with our CF-VAE (Figure 3). This can be seen in the samples and corresponding latent spaces in Figure 9.

**Hyper-parameter analysis of our posterior regularization scheme (pR).** We provide additional analysis of our posterior regularization scheme in Table 5. We observe that our CF-VAE is relatively robust across $C = [0.05, 025]$, with only small variance in performance. This is because our posterior regularization scheme encourages our CF-VAE to focus on explaining the data well. We explain this further in the following paragraph.

**Analysis of our posterior regularization scheme (pR).** We provide additional analysis of our posterior regularization scheme (pR) in Figure 10. We show each term of our objective (7) in Figure 10. First, we see that with our posterior regularization scheme, our CF-VAE focuses on explaining the data well – the data log-likelihood is best with our posterior regularization (pR) scheme Figure 10a, with $C = 0.2$ having a advantage over $C = \{0.05, 0.1\}$. Furthermore, we see that without our posterior regularization scheme the Jacobian term dominates while entropy term decreases (Figure 10b vs Figure 10d) – the contraction of the base density is favoured. Interestingly,

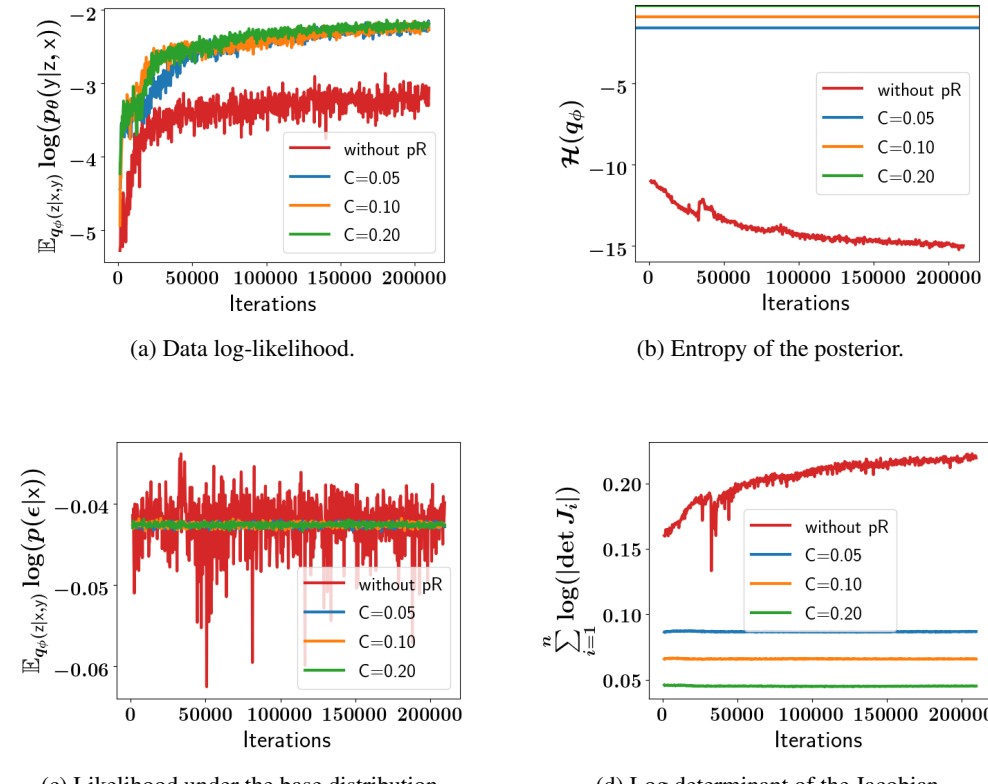

(a) Data log-likelihood.

(b) Entropy of the posterior.

(c) Likelihood under the base distribution.

(d) Log determinant of the Jacobian.

Figure 10: Analysis of all four terms of our CF-VAE objective (7) at training time, with ($C = \{0.05, 0.10, 0.20\}$) and without our posterior regularization (pR) scheme. We observe better data log-likelihoods and stable training with our posterior regularization (pR) scheme. Without pR, we observe that the Jacobian term dominates at the cost of data log-likelihood.

the likelihood under the prior Figure 10c is similar across methods – with our posterior regularization providing additional stability. We also experimented with re-weighting these terms (although its no longer a valid lower bound on the true data log-likelihood). This leads to the opposite behaviour – the entropy term dominates over the Jacobian term at the cost of the data log-likelihood. On the other hand, we observe that all terms of our objective are stable with our posterior regularization scheme, illustrating the advantage of our posterior regularization scheme.

## APPENDIX F. EVALUATION OF THE ROBUSTNESS OF THE TOP N% METRIC

We use two simpler uniform "Shotgun" baselines to study the robustness of the Top n% metric against random guessing. In particular, we consider the "Shotgun"-u90° and "Shotgun"-u135° baselines which: given a budget of N predictions, it uniformly distributes the predictions between $(-90°, 90°)$ and $(-135°, 135°)$ respectively of the original orientation and using the velocity of the last time-step. In Table 6 we compare the Top 1 (best guess) to Top 10% metric with N= $50, 100, 500$ predictions.

We see that in case of both the "Shotgun"-u90° and "Shotgun"-u135° baselines, the Top 1 (best guess) metric improves with increasing number of guesses. This effect is even more pronounced in case of the "Shotgun"-u135° baseline as the random guesses are distributed over a larger spatial range. In contrast, the Top 10% metric remains remarkably stable. This is because, in order to improve the Top 10% metric, random guessing is not enough – the predictions have to be on the correct modes. In other words, the only way to improve the Top 10% metric is move random predictions to any of the correct modes.

| Method | K | Error @ 1sec | Error @ 2sec | Error @ 3sec | Error @ 4sec |
|---|---|---|---|---|---|
| | | Top 1 (Best Guess) | | | |
| "Shotgun"-u90° | 50 | 0.9 | 1.9 | 3.1 | 4.4 |
| "Shotgun"-u90° | 100 | 0.9 | 1.9 | 3.0 | 4.3 |
| "Shotgun"-u90° | 500 | 0.9 | 1.9 | 3.0 | 4.3 |
| | | Top 10% | | | |
| "Shotgun"-u90° | 50 | 1.2 | 2.5 | 3.9 | 5.4 |
| "Shotgun"-u90° | 100 | 1.2 | 2.5 | 3.9 | 5.4 |
| "Shotgun"-u90° | 500 | 1.2 | 2.5 | 3.9 | 5.4 |
| | | Top 1 (Best Guess) | | | |
| "Shotgun"-u135° | 50 | 0.9 | 2.0 | 3.1 | 4.5 |
| "Shotgun"-u135° | 100 | 0.9 | 1.9 | 3.0 | 4.3 |
| "Shotgun"-u135° | 500 | 0.9 | 1.9 | 3.0 | 4.2 |
| | | Top 10% | | | |
| "Shotgun"-u135° | 50 | 1.4 | 2.9 | 4.5 | 6.2 |
| "Shotgun"-u135° | 100 | 1.4 | 2.9 | 4.5 | 6.2 |
| "Shotgun"-u135° | 500 | 1.4 | 2.9 | 4.5 | 6.2 |

Table 6: Five fold cross validation on the Stanford Drone dataset. Euclidean error at ($1/5$) resolution.

## APPENDIX G. QUALITATIVE EXAMPLES ON THE HIGHD DATASET

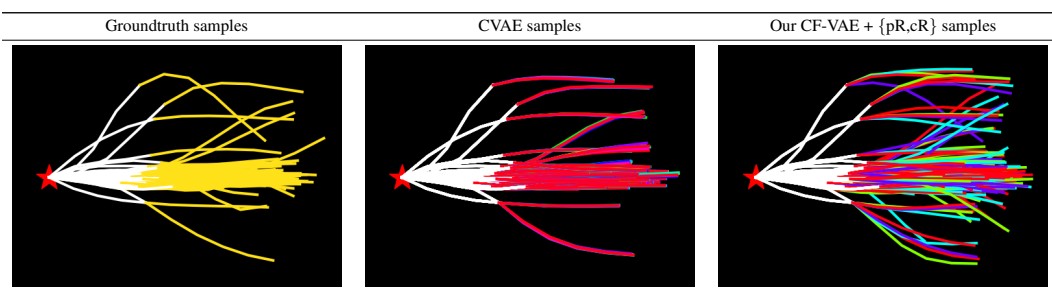

Figure 11: Predictions on the HighD dataset. Left: 128 random samples from the HighD test set (in yellow). Middle: CVAE predictions (5 samples per test set example). Right: Our CV-VAE + {pR,cR} predictions (5 samples per test set example). While the predictions by the CVAE are linear continuations, our CF-VAE sample predictions are much more diverse and cover events like lane changes e.g. top most sample track from the test set.

We show qualitative examples on the HighD dataset in Figure 11. In the left of Figure 11 we show 128 random samples from the HighD test set. In the middle we show predictions on these samples by the CVAE (with cyclic KL annealing (Liu et al., 2019)). We see that even with cyclic KL annealing, we observe posterior collapse. All samples have been pushed towards the mean and the variance in the 5 samples per test set example is minimal. E.g. note the top most sample track from the test set in Figure 11 (left). All CVAE sample predictions are a linear continuation of the trajectory (continuing on the same lane), while there is in fact a turn (change of lanes). In contrast, our CF-VAE + {pR,cR} sample predictions are much more diverse and cover such eventualities. This also shows that our CF-VAE + {pR,cR} does not suffer from such posterior variable collapse.

