# OpenReview forum: "Conditional Flow Variational Autoencoders for Structured Sequence Prediction"
_ICLR.cc/2020/Conference — Reject_

### Official Review · AnonReviewer3 · 2019-10-22
**Official Blind Review #3**

**Rating:** 6

**Review:**

The paper proposes a combination of conditional VAE wtih normalising flows priors and posterior regularisation strategies to capture the diversity of multi-modal trajectories of complex motion patterns. The paper argues that more flexible priors over the latent space can provide posteriors that more closely resemble the trajectories observed in the training data. To this end, the paper presents a derivation of the evidence lower bound for VAEs with normalising flows and discusses the effect of fixing the variance of the posterior to reduce instability during training. Additionally, it shows that conditioning the regularisation on whether or not the dataset contains a dominating mode leads to more diversity and captures minor modes more effectively. Experiments are reported on sequence datasets of handwritten digits, and two datasets with trajectories of vehicles in traffic.

A central point the paper makes is the importance of prior distributions for the latent space in VAEs such that it can capture diverse modes of trajectories. It is well known that more flexible priors such as MoG lead to better generative power as shown in Tomczak and Welling, 2018. The paper focuses on an extension of the work by Ziegler and Rush, 2019 which proposes normalising flows as priors to capture sequences, conditioned on the initial part of the trajectory. This extension is relatively simple, but does address the specifics of the problem well.

In general, the paper is well written and clear. The main innovation, in my opinion, is the combination of several ideas applied to the problem of sequence prediction. While none of the ideas, in isolation, are significantly new, the combination can be useful to this particular problem. However, I would like feedback from the authors on the following two main points below which are the main weaknesses of the paper:

1) Posterior regularisation: The posterior regularisation strategies, while intuitive, are very ad-hoc and somewhat contrary to the Bayesian framework. It is difficult to see how the variance in pR and the "dominant mode" detector in cR can be estimated automatically. Within a Bayesian framework it would be much more natural to place a prior distribution over the variance and marginalise it out within the variational inference procedure. For the other regularisation (cR), how is the dominant mode detected?

2) Experiments: A major concern reported throughout the paper is the instability of training and the risk for overfitting. I do not think the experiments demonstrate how stable and robust the method is to different initialisations, seeds, training data shuffles, etc. I strongly suggest the authors to run cross validation experiments and report the mean and standard deviation for all methods being compared. Also, how sensitive are the results to different values of C? How to decide whether to use cR or not when don't have access to the ground truth?

**Experience Assessment:**

I have published one or two papers in this area.

**Review Assessment: Checking Correctness Of Derivations And Theory:**

I assessed the sensibility of the derivations and theory.

**Review Assessment: Checking Correctness Of Experiments:**

I assessed the sensibility of the experiments.

**Review Assessment: Thoroughness In Paper Reading:**

I read the paper thoroughly.

---

> ### Author Response · Authors · 2019-11-10
> **Response to Review #3 - 1/1**
>
> We thank the reviewer for the comments and address them here in detail.
>
> * ‘ the paper is well written and clear’ - Thank you.
>
> * ‘While none of the ideas, in isolation, are significantly new, the combination can be useful to this particular problem’ - We thank you for recognizing the significance of our approach. Please also note that we propose the first conditional normalizing flow based model for structured sequence prediction, the first conditional non-linear flows along with two new regularization schemes.
>
> * ‘estimating variance in pR’ - As intuition would suggest, reasonably small values of C work well -- as they allow for good data reconstruction and also makes it easy for our conditional flow to fit the marginal posterior. As shown in Table 5 and  Figure 10 of Appendix E, our model is robust across a large range of C=[0.05,0.25]. Furthermore, we observe that these values are robust across all three datasets - MNIST Sequences, Stanford Drone and HighD. Therefore, we did not face any challenges in estimating the variance of pR.
>
> * ‘"dominant mode" detector in cR ’ - Please note that, we do have to directly detect the dominant mode for our condition regularization scheme. The dominant mode is the mode which dominates (explains a large part) the data log-likelihood term. E.g. in case of HighD we observe that ~90% of the data log-likelihood is dominated by the main mode which is the case of the vehicle moving straight on the highway without a lane change. Posterior collapse occurs in case of conditional latent variable models because the model focuses on this dominant mode and chooses to ignore the latent variables (and thus the other minor modes)  because this leads to a easier to encode latent distribution. Our condition regularization scheme encourages the model to focus on all modes by ensuring that the latent variables cannot be ignored.
>
> * ‘prior distribution over the variance’ - This is an interesting idea. However, it is not straightforward to implement. This is because this would require us to enforce a prior over the Jacobian of our conditional flow prior (as the variance of the posterior is dependent on the Jacobian). It would be challenging to enforce such a prior without affecting the expressivity of our conditional flows. In contrast, our posterior regularization scheme is straightforward to implement, robust and leads to state-of-the-art results.
>
> * ‘demonstrating the stability and robustness of the method’ - We have added additional experiments in Table 5 and Figure 10 of Appendix E to further illustrate the stability and robustness of the method. Please note, the results in the main paper are the mean of five independent runs with random initializations. We additionally report the standard deviation of 5 runs in Table 5, across all baselines. We observe low standard deviation across runs demonstrating the stability of our method. Furthermore, we also observe stable performance across values a large range of the posterior regularization hyper-parameter C=[0.05,0.25].
>
>
> * ‘Data shuffles and overfitting’ - We report results on standard the MNIST Sequence and HighD test set as in prior work. Furthermore, we report 5-fold cross validation results on Stanford Drone in Table 2 (following prior) work. These results demonstrate that our method is effective across data shuffles and does not suffer from overfitting.
>
>
> * ‘How to decide whether to use cR’ - We find that in practice we this can be decided on the basis of the training data. E.g. the training sequences can be clustered (with k-means) to determine if there is a dominant mode. E.g. in case of HighD k-means clustering reveals that the dominant mode is the case where the car continues travelling straight along the highway.
>
> Finally, we thank the reviewer for voicing her/his concerns and helping us improve our work.  We would be happy to answer any remaining questions.

---

### Official Review · AnonReviewer1 · 2019-10-23
**Official Blind Review #1**

**Rating:** 6

**Review:**

The work proposes a method to improve conditional VAE with a learnable prior distribution using normalizing flow. The authors also design two regularization methods for the CF-VAE to improve training stability and avoid posterior collapse. The paper is clearly motivated and easy to follow. Experiment results on MNIST, Stanford Drone and HighD datasets show the proposed that the model achieves better results than previous state-of-the-art models by significant margins.

However, the reviewer has the following comments on improving the paper:

The motivation of the conditional normalizing flow design could be made more clear. The posterior regularization originates from the problem that the log Jacobian term encourages contraction of the base distribution. The log Jacobian term would be zero and would not encourage the contraction of the base distribution if the normalizing flow was volume-preserving, like NICE (http://proceedings.mlr.press/v37/rezende15.pdf, https://arxiv.org/pdf/1410.8516.pdf), which could be to convert into a conditional normalizing flow. On the MNIST results, the CF-VAE model with the proposed conditional normalizing flow even has worse performance than the affine flow model without the regularization. Therefore, clarifying the motivation behind this design choice is important.

The work claims the two regularization methods are used to avoid a low-entropy prior and posterior collapse. But the claims are not fully substantiated in the experimental results. It would be better if the paper explicitly compares the CF-VAE models with and without regularizations in terms of the entropy of prior distribution and KL divergence.

**Experience Assessment:**

I have read many papers in this area.

**Review Assessment: Checking Correctness Of Derivations And Theory:**

N/A

**Review Assessment: Checking Correctness Of Experiments:**

I assessed the sensibility of the experiments.

**Review Assessment: Thoroughness In Paper Reading:**

I read the paper at least twice and used my best judgement in assessing the paper.

---

> ### Author Response · Authors · 2019-11-10
> **Response to Review #1 - 1/1**
>
> We thank the reviewer for the comments and address them here in detail.
>
> * ‘The paper is clearly motivated and easy to follow.’ - Thank you.
>
> * ‘Experiment results on MNIST, Stanford Drone and HighD datasets show the proposed that the model achieves better results than previous state-of-the-art models by significant margins.’ - Thank you.
>
> * ‘Volume-preserving flows’ - We have added results with the volume preserving NICE flows in Table 5 in Appendix E. We observe that even without our posterior regularization scheme (pR) non-volume preserving NICE flows (Dinh et. al. 2015) performs well -- because of the constant Jacobian term. However, our conditional non-linear flows with posterior regularization still perform significantly better (78.9 vs 74.9 -CLL). This is because of the additional expressive power of our conditional non-linear flows combined with the stability offered by our posterior regularization scheme. Please note that the results with Affine flows in Table 1 already includes posterior regularization. We apologize for not pointing this out in the manuscript. We have updated our manuscript to reflect this.
>
> * ‘comparing the CF-VAE models with and without regularizations’  - We have added additional results in Figure 10 of Appendix E illustrating the effect of our posterior regularization scheme on each of the four terms of our objective, 1. The data log-likelihood, 2.  The entropy of the posterior, 3. The log-likelihood under the base Gaussian distribution of the conditional flow prior, 4. The log-determinant of the Jacobian. First, we see that with our posterior regularization scheme, our CF-VAE focuses on explaining the data well -- the data log-likelihood is best with our posterior regularization (pR) scheme, with C=0.2 having an advantage over C={0.05,0.1}. Furthermore, we see that the Jacobian term of our conditional flow dominates while entropy term decreases -- the contraction of the base density is favoured. We also experimented with re-weighting these terms (although it's no longer a valid lower bound on the true data log-likelihood). This lead to the opposite behaviour -- the entropy term dominates over the Jacobian term at the cost of the data log-likelihood. On the other hand, we observe that all terms of our objective are stable with our posterior regularization scheme, illustrating the advantage of our posterior regularization scheme.
>
> Finally, we thank the reviewer for voicing her/his concerns and helping us improve our work.  We would be happy to answer any remaining questions.

---

### Official Review · AnonReviewer2 · 2019-10-23
**Official Blind Review #2**

**Rating:** 1

**Review:**

The paper demonstrates how normalising flows can be conditioned. The method is then demonstrated on a set of sequential experiments which show improvements over the considered base lines.

I recommend rejection of the paper, but I can see me changing that assessment if certain improvements are made. The central points are:
- the paper has errors,
- the paper does not respect some related work and has been published previously in parts,
- the paper has a claim that is unsupported in my view,
- the paper is overcrowded with annoying marketing language; the word "novel" appears 16 times according to my pdf viewer.

In general I like the idea, and the presentation seems solid to a large degree. However, the above points are a show stopper for me personally.

For one, the statements

- p(y|x) = p(y|x, z) p(z | x) and
- p(y|x) = p(y|z) p(z|x),

are problematic. I would like the authors to clarify how they arrive at these.

The paper starts with the claim that "prior work [...] imposes a uni-modal standard Gaussian prior on the lagent variables". This is just wrong. The whole literature of stochastic recurrent models does not do this. See  [1, 2] for starting points. Since the authors place their work in the setup of sequential prediction, this is what has to be respected.

Further, the authors do not seem to be aware of a recently published work [3] that adresses *exactly* this problem. To quote from their abstract: "To this end, we modify the latent variable model by defining the likelihood as a function of the latent vari- able only and [sic] introduce an expressive multimodal prior to enable the model for capturing semantically meaningful features of the data."

I have two more questions with respect to the proposed regularisations.

First, I would ask the authors to comment on the relationship of cR and the method proposed in [3]. To me, it appears as if cR is not novel, but has instead been proposed in [3] previously.

Second, pR fixes the variance of q. The authors claim that the normalising flow of the conditional can undo this fixing by adequately scaling the prior. Hence, so the claim, the expressivity of the model is not reduced.

This prohibits the posteriors of two distinct data points to share the same mean but not share the same variance.

I request the authors to make a more formal analysis of this, as I do am not convinced how the expressivity of the model is maintained and what influence this has on the ELBO.



References
[1] Bayer, Justin, and Christian Osendorfer. "Learning stochastic recurrent networks." arXiv preprint arXiv:1411.7610 (2014).
[2] Chung, Junyoung, et al. "A recurrent latent variable model for sequential data." Advances in neural information processing systems. 2015.
[3] Klushyn, Alexej, et al. "Increasing the Generalisaton Capacity of Conditional VAEs." International Conference on Artificial Neural Networks. Springer, Cham, 2019.

**Experience Assessment:**

I have published in this field for several years.

**Review Assessment: Checking Correctness Of Derivations And Theory:**

I carefully checked the derivations and theory.

**Review Assessment: Checking Correctness Of Experiments:**

I assessed the sensibility of the experiments.

**Review Assessment: Thoroughness In Paper Reading:**

I read the paper thoroughly.

---

> ### Author Response · Authors · 2019-11-10
> **Response to Review #2 - 2/2**
>
> * ‘the expressivity of the model is not reduced’ - We apologize for the  inclarity. Here, by expressivity we refer to whether the marginal posterior distribution of latents q_{\phi}(z|x) is expressive enough to explain the data and whether the prior can match the posterior -- demonstrated by sample quality at test time. We have updated the text. To better analyze this we provide two additional sets of experiments in Table 5 and Figure 10 in Appendix E. We show the data log-likelihood during training for various values of C and without pR, 2. We additionally report the test log-likelihood for the corresponding values of C. Note that, with pR (fixed C) the ELBO would always be less than or equal to the ELBO without pR. However (irrespective of the total value of the ELBO), we see that we consistently obtain better data log-likelihoods during training with pR in Figure 10 (a). As mentioned in the manuscript, a constant value of C encourages our model to concentrate on explaining the data. The objective without posterior regularization is dominated by the Jacobian at the cost of the data log-likelihood (Figure 10 (b) vs Figure 10 (d)) -- while the likelihood under the base distribution is identical. This is further illustrated by our test log-likelihoods which show that our conditional flow prior can scale to deal with different values of fixed C=[0.05,0.30] and thus leads to better sample quality at test time (also see Figure 3).
>
> * We apologize for our writing style. We tried to clearly present and highlight our contributions. We have tried to improve our writing style in the current draft.
>
> Finally, we thank the reviewer for voicing her/his concerns and helping us improve our work.  We would be happy to answer any remaining questions.

---

> ### Author Response · Authors · 2019-11-10
> **Response to Review #2 - 1/2**
>
> We thank the reviewer for the comments and address them here in detail.
>
> * ‘In general I like the idea, and the presentation seems solid to a large degree.’ - Thank you.
>
>
> * ‘the statements p(y|x) = p(y|x, z) p(z | x) and p(y|x) = p(y|z) p(z|x)’ - We thank you for pointing these out these typos. To clarify, these statements are missing the integral over z, e.g. p(y|x) = \int p(y|x, z) p(z | x) dz. Additionally, regrading the second statement, please note that we assume a strong conditional normalising flow based prior that can encode conditioning information in the latent space such that p(y|x,z) = p(y|z). We have updated the text to reflect this.
>
> * ‘prior work [...] imposes a uni-modal standard Gaussian prior’ - We apologize for this inclarity and we thank you for pointing out [1,2]. We have updated the manuscript (including the abstract) and included these references. However, there seems to be a misunderstanding here.  First, we have included extensive references to prior work on expressive priors in the introduction and related work section. Secondly, please note that [1,2] uses sequential latent variables - a latent variable is sampled at every time-step. Our CF-VAE (following prior work e.g. Lee et. al.,2017; Bhattacharyya et. al., 2018) samples a global latent variable for prediction of the entire future sequence. The references [1,2] do impose uni-modal Gaussian priors at each time-step. Please refer to page 4 of [1] which states “In this work, we restrict ourselves to a standard Normal prior”. Similarly, Equation (5) of [2] states the same. Therefore, we believe that there are significant differences between [1,2] and our work.
>
> * ‘recently published work [3]’ - Please note, this work [3] was submitted to arXiv on 23rd August, 2019 (https://arxiv.org/abs/1908.08750). Furthermore, the proceedings were published (to the best of our knowledge) in September 2019 (https://e-nns.org/icann2019/) -- and the content is behind a paywall. Our work was submitted to arXiv on the 24th of August, 2019 (can be independently verified. Please also note that ICLR does allow submission to arXiv). Also please note, ICLR (https://iclr.cc/Conferences/2019/Reviewer_Guidelines - there is no updated version for 2020) has the policy - “no paper will be considered prior work if it appeared on arxiv, or another online venue, less than 30 days prior to the ICLR deadline.'' Therefore, following the ICLR policy, we consider this as parallel work.
>
> However, we found the work [3] very interesting. We have added [3] as a reference in our manuscript and added a discussion. We believe that the main difference between [3] and our condition regularization scheme is that  we employ this regularization to deal with posterior collapse only in case of distributions with dominant modes. We do not always need this regularization to learn rich latent spaces e.g. in case of MNIST Sequences and Stanford Drone datasets. We also found the proposed CDV prior in [3] very interesting. Therefore, we include additional experiments with the proposed CDV prior of [3] in Appendix E.
>
> Please also consider that the condition regularization scheme is not our main contribution. We propose the first conditional normalizing flow based model for structured sequence prediction, the first conditional non-linear flows along with the posterior regularization scheme. Therefore, we believe that our work is significantly distinct from [3].

---

### Decision · Program_Chairs · 2019-12-19

**Decision:**

Reject

**Comment:**

The novelty of the proposed work is a very weak factor, the idea has been explored in various forms in previous work.